# Semi-Supervised Domain Adaptation for Multi-Label Classification on Nonintrusive Load Monitoring

**DOI:** 10.3390/s22155838

**Published:** 2022-08-04

**Authors:** Cheong-Hwan Hur, Han-Eum Lee, Young-Joo Kim, Sang-Gil Kang

**Affiliations:** 1Department of Computer Engineering, Inha University, Inha-ro 100, Nam-gu, Incheon 22212, Korea; 2Electronics and Telecommunications Research Institute (ETRI), 218 Gajeong-ro, Yuseong-gu, Daejeon 34129, Korea

**Keywords:** nonintrusive load monitoring, transfer learning, domain adaptation, pseudo labeling, semi-supervised learning, appliance usage classification

## Abstract

Nonintrusive load monitoring (NILM) is a technology that analyzes the load consumption and usage of an appliance from the total load. NILM is becoming increasingly important because residential and commercial power consumption account for about 60% of global energy consumption. Deep neural network-based NILM studies have increased rapidly as hardware computation costs have decreased. A significant amount of labeled data is required to train deep neural networks. However, installing smart meters on each appliance of all households for data collection requires the cost of geometric series. Therefore, it is urgent to detect whether the appliance is used from the total load without installing a separate smart meter. In other words, domain adaptation research, which can interpret the huge complexity of data and generalize information from various environments, has become a major challenge for NILM. In this research, we optimize domain adaptation by employing techniques such as robust knowledge distillation based on teacher–student structure, reduced complexity of feature distribution based on gkMMD, TCN-based feature extraction, and pseudo-labeling-based domain stabilization. In the experiments, we down-sample the UK-DALE and REDD datasets as in the real environment, and then verify the proposed model in various cases and discuss the results.

## 1. Introduction

Understanding energy usage in buildings has been considered an important issue because residential and commercial power consumption account for about 60% of global energy consumption [1]. Optimized energy usage management has advantages for both suppliers and consumers of energy. From the supplier’s point of view, planned consumption may be encouraged according to the frequency and pattern of use of home appliances. In addition, it is also easy for consumers to develop plans that can reduce costs through comprehensive information about device-specific operations [2]. The electricity usage profile is to install a submeter for each appliance and record instantaneous power readings, but in reality, applying this method to all devices is difficult to realize due to cost and difficulty in maintenance. Therefore, nonintrusive load monitoring (NILM) aims to disaggregate energy consumption by device. The NILM method that does not depend on submeters has shown significant efficiency in commercial and residential energy utilization and remains an important task [3].

NILM is inherently difficult because it analyzes information about the simultaneous switching or noise generation of multiple devices without attaching multiple submeters [4,5,6]. To solve the problem, many techniques such as dynamic time warping (DTW), matrix factorization, neuro-fuzzy modeling, and graph signal processing (GSP) have been proposed and supervised and unsupervised learning-based techniques have been studied [7,8,9]. Hart [10] first introduced unsupervised learning methods to decompose electrical loads through clustering. However, various techniques such as hidden Markov models (HMM) have been proposed for a while because clustering-based methods do not have training data and are difficult to predict accurate power loads. In recent years, the number of research studies on deep neural networks (DNNs) has increased rapidly with the advancement of high-end hardware devices, and the availability of data for supervised learning has increased [11]. Long short-term memory (LSTM), a representative supervised learning technology, considered NILM as a prediction problem based on time series data. Refs. [12,13] proposed a method for learning models by controlling data applied with various data sampling-based windows. Nolasco et al. [14] included multi-label procedures to increase the recognition rate for multi-loads by marking on loads at any given time and developed architectures based on convolutional neural networks (CNNs), resulting in an outstanding performance in signal detection and feature extraction. However, existing supervised learning methods for NILM still have two problems. First, there is a fundamental problem that assumes that the power usage data of real devices has a distribution similar to that of training data. It is impossible to ensure the same performance in actual situations because devices of the same type have different energy consumption depending on products and brands, noise form, intensity, physical environment, etc. [15]. To overcome this problem, training data containing all domain information must be acquired, but it is practically impossible since collecting the energy consumption of each device from different houses requires huge costs. Another problem is that, even assuming that neural network models are trained on all the data for different environments, extracting critical information is very difficult because of the vast amount of complex data [16,17,18]. Therefore, identifying suitable techniques that can handle the large complexity of data and generalize various domains of information is the main challenge in NILM.

To solve these problems, we consider domain adaptation [19,20]. Domain adaptation is one of the transfer learnings, which can adapt the trained model to the other domain dataset on the same task. This concept can easily be applied to the NILM system. Many researchers proposed domain adaptation systems to generalize various domain information [21,22]. Liu et al. [21] conducted a regression study to refine energy consumption by applying the most typical domain adaptation method to NILM. Since only the basic concept of domain adaptation has been applied to NILM, it has the potential to develop in various ways. Ref. [22] proposed a method that incorporates the mean teacher method into domain adaptation. Regression work is performed on the source and target domains using one model. However, this method did not show good performance in domain generalization due to its shallow model structure. To the best of our knowledge, there are no papers on classification tasks in domain adaptation studies for NILM. In this paper, we perform classification tasks for device usage detection in NILM by incorporating powerful feature information distillation based on the teacher–student structure and pseudo-labeling (PL) into domain adaptation.

The main contents of this paper are as follows:We conduct the first classification study in the domain adaptation field for NILM;We show performance improvements by incorporating robust feature information distillation techniques based on the teacher–student structure into domain adaptation;The decision boundaries are refined through PL-based domain stabilization.

The remainder of this paper is organized as follows. Section 2 shows a brief review of related studies of NILM and domain adaptation. Section 3 introduces the proposed method. Section 4 presents the experimental setup, case study, and discussions. Finally, Section 5 concludes the paper.

## 2. Related Work

### 2.1. Nonintrusive Load Monitoring

Consider a building with m appliances and k operating power modes of each appliance for time $\left[1,\;\ldots ,T\right]$. Let $x_{i}=(x_{i}\left(1\ldots \ldots ,x_{i}\left(T\right)\right)$ denote the energy consumption of the i-th device. The whole energy usage of the i-th device in sample time n can be formulated as follows:(1)$$x_{i}\left(n\right)=\left[U_{1}^{i}\ldots ),\;\ldots ,U_{k}^{i}\left(n\right)\right]\left[\begin{array}{c} \psi_{1}^{i}\\ \begin{array}{c} \vdots \\ \psi_{k}^{i} \end{array} \end{array}\right]+\epsilon^{i}\left(n\right)$$
where $\psi_{k}^{i}$ is the electricity consumption consumed in a particular operating mode, $\epsilon^{i}\left(n\right)$ denotes the measurement of background noises, and $U_{k}^{i}\left(n\right)$ is the operating On/Off $\left[0,1\right]$ status of the i-th appliance in time n. The operating status assures the equality constraint $\sum_{j=1}^{k}U_{j}^{i}\left(n\right)=1$ since all appliances operate in a single mode. At time *n*, the final energy aggregate of the house is expressed as follows:(2)$$x\left(n\right)=\sum_{i=1}^{m}\left[\ldots \left(n\right),\;\ldots ,U_{k}^{i}\left(n\right)\right]\left[\begin{array}{c} \psi_{1}^{i}\\ \begin{array}{c} \vdots \\ \psi_{k}^{i} \end{array} \end{array}\right]+\epsilon^{i}\left(n\right)$$

The goal of the NILM algorithm is to disaggregate the measured electricity usage x to generate appliance-specific energy consumption profiles [23,24]. Therefore, the final challenge is to reduce the difference between the actual measurements of the device and the disaggregated energy consumption [25]. 

Elafoudi et al. [7] detected the edge within the time window and used DTW to identify the unique load signature. Lin et al. [8] proposed a hybrid classification technique that combines fuzzy c-means and clustering piloting with neuro-fuzzy classification to distinguish devices that have similar load signatures. He et al. [9] handled the NILM as a single-channel blind source separation problem to perform low-complexity classification active power measurements. Based on this idea, they proposed the GSP-based NILM system to handle the large training overhead and the computational cost of the conventional graph-based method.

### 2.2. Domain Adaptation

Domain adaptation is an area of transfer learning [26]. In general transfer learning, a task or domain can be changed from source to target; however, in domain adaptation, the task sets the premise that only the domain is changed [19,27]. This aims to generalize the classification or regression model, which is trained on the source domain to be applied to the target domains with different distributions, since distribution disagreement between training and real data yields poor model performance. Ganin et al. [19] proposed a multi-task learning model with a class and domain classifier. The model was trained to only classify class labels, except for domain labels. For this, they introduced the gradient reversal layer (GRL) to the domain classifier. The GRL multiplies the negative constant and the gradient on the backward pass. Additionally, it makes the model remove the domain information in its feature extractor. With the advancement of deep neural networks (DNNs), the performance of domain adaptation has achieved outstanding performance in various fields [11,14,28,29,30,31,32,33]. In [34], domain adversarial training of neural networks (DANN), inspired by the generative adversarial network (GAN), laid the foundation for applying adversarial learning methodologies to domain adaptation and accomplished excellent performance. In addition, domain adaptation algorithms based on maximum mean discrepancy (MMD) between source and target were mainly studied [35,36,37,38]. In [39], Long et al. proposed a joint MMD to adjust the joint distribution. Deep domain confusion (DDC) [34] proposed a technique for using pre-trained networks by adding adaptive layers based on MMD. 

Although domain adaptation is used in various fields as expressed above, the application of domain adaptation in NILM has not been researched a lot and requires advancement. In [40], Wan proposed a domain adaptation algorithm for optical character recognition (OCR), which was extended to apply it to the NILM field and produce prominent results. Recently, Wang and Mao proposed applying a model-agnostic meta-learning (MAML)-based domain adaptation algorithm to NILM, inspired by the pre-trained model, which is heavily studied in the NLP field and outperformed the state-of-the-art deep learning-based methods. [41].

## 3. Semi-Supervised Domain Adaptation for Multi-Label Classification on Non-Intrusive Load Monitoring

Various deep learning models are applied to the NILM field. However, the task of segmenting the use of different devices in many houses is still a relatively new concept. To solve this problem, we propose the semi-supervised domain adaptation for multi-label classification on non-intrusive load monitoring. The overall diagram is shown in Figure 1.

Several hypotheses are proposed in this work to apply semi-supervised domain adaptation to NILM. The first hypothesis is that the distribution of source and target domains is different. Most NILM systems are based on this hypothesis. We also use labels on the source for domain adaptation, not on the target. Second, even if the distribution of source domain data and target domain data is different in NILM, it is assumed that the same device has domain-independent common characteristics regardless of the domain. Because in motor devices, lagging current with slow current flow occurs, which results in a low power factor. Additionally, capacitor devices generate leading current with fast current flow, which results in a high-power factor. The power factor is the ratio of active power to apparent power regardless of the magnitude of power consumption. In other words, if two different houses use the same electronic devices (e.g., refrigerator, TV, etc.) from different manufacturers, it is assumed that there is a common usage pattern even if the power consumption is different.

The proposed method consists of three main steps, shown in Figure 2. In the knowledge distillation stage, high-level knowledge is distilled into the student network (SN) by a temporal convolutional network (TCN) [42]-based teacher network (TN) [43] trained using labeled source data. Domain-dependent features vary depending on the domain, and domain-independent features are constant regardless of the domain. In the next step, we perform a robust domain adaptation that allows us to extract only domain-independent features to adapt source and target data to neural networks regardless of domain. Appliance usage detection classifies devices from source domain data. Additionally, domain classifiers are trained with GRL to prevent classification for source and target domains. As a result, feature extractors can extract robust domain-independent features that enable device usage classification regardless of domain. In the domain stabilization step in Figure 2, we stabilize the domain through PL-based fine-tuning. First, domain-independent features of target data are extracted from the feature extractor and then pseudo-labeled based on the source domain label in appliance usage detection. Since all the target data cannot be pseudo-labeled, it is partially pseudo-labeled. Therefore, the target data consists of pseudo-labeled data and unlabeled data. Secondary domain adaptation is performed based on the enhanced target domain data and domain-independent features extracted through robust distillation. The network performance is stabilized and improved through the advantages of low-density separation between classes and entropy regularization. Details of each part of the proposed framework are in the subsections.

### 3.1. Network Architecture

The goal of this section is to build a semi-supervised domain adaptation model that can estimate the target domain label $Y_{t}$ using labeled source data $\left(X_{s},\;Y_{s}\right)$ and target data $X_{t}$. As shown in Figure 2, the model includes three parts: knowledge distillation, robust domain adaptation, and domain stabilization. Details of the network structure are as follows:

(1)Knowledge distillation: knowledge is distilled using a TCN feature extraction-based teacher–student network to receive robust domain-independent features of source data. TCN is an extended time-series data modeling structure in CNN. It provides better performance than typical time-series deep learning models such as LSTM because it has a much longer and more effective memory without a gate mechanism. The TCN consists of several residual blocks, and this block consists of a dilated casual convolution operation O. For input x∈ℝn and filter ft :0,1…,k−1→ℝ, O at point s is defined by Equation (3).
(3)Os=x∗dfts=∑i=0k−1fti·xs−d·i
where d is the dilated factor, ∗d is ∗d-dilated convolution, k is the filter size, and s−d·i is the past value. However, as the network depth increases, performance decreases rapidly due to overfitting. However, as the network depth increases, performance decreases quickly due to overfitting. Resnet’s key concept, namely, residual mapping, can solve this problem. The TCN residual block includes two layers of dilated casual convolution based on the ReLU activation function, weighted normalization, and dropout. The 1 × 1 convolution layer on the TCN ensures that the input and output are the same size. The output of the transformation T of the time series data in the TCN dual block is added to the identity mapping of the input x and expressed as follows:(4)Rs=Tx,θ+x
where θ means the set of parameters of the network. It has already been demonstrated that this concept of residual block improves network performance by learning modifications to identity mapping rather than overall transformations. Based on this, it is possible to build a deep TCN network by stacking multiple TCN residual blocks. Assuming that xI is the input of the I-th block, the network forward propagation from the I-th block to the I+n-th block can then be formulated as follows:(5)xI+n=xI+∑i=II+n−1Txi,θi
where, xI is I-th block, θi is the parameter set of the i-th block. Therefore, the feature extractor FEtexs, θfte of the TN is defined as follows:(6)FEtexs, θf_te=xs+∑i=0k−1Txs_i,θf_te_i
where the number of layers is k, xs is source data, θfte is the parameter set of TN, xsi is ith source data, θftei is the parameter set of ith block in the TN. Additionally, the feature extractor FEstxs, θfst of SN can be defined as follows:(7)FEstxs, θf_st=xs+∑i=0l−1Txs_i,θf_st_i
where l is the number of layers, θfst is the parameter set of SN, θfsti is the parameter set of ith block in SN. Based on fte extracted from Equation (6), the TN must extract soft label information for transferring knowledge to the SN through appliance usage detection, which consists of a fully connected layer. The output y^te of the TN is defined as follows:(8)y^te=Softmaxwith TAUDtefte, θtei=eAUDtefte, θteiT∑j=1KeAUDtefte, θtejT
where te refers to the TN, y^te is a predicted classification label of xs in the TN, T is a temperature parameter, Softmaxwith T is a Softmax function with a temperature parameter. θte is the parameter set of AUDte, AUDtefte, θtei is the elements of output vector of AUDte, i refers to ith element, K is the number of elements of the output vector. Maximize the benefits of soft label values for knowledge distillation by using temperature parameters to prevent information loss in Softmax output. The estimated soft label y^te is compared to the soft prediction y^stsp of SN and is used as a distillation loss in network training. y^stsp is obtained as follows:(9)y^st_sp=Softmaxwith TAUDstfst_s, θsti=eAUDstfst_s, θstiT∑j=1KeAUDstfst_s, θstjT
where st refers to SN, y^stsp is a predicted classification label of xs in the SN and a soft prediction value of SN, θst is the parameter set of AUDst, and AUDstfsts, θsti is the i-th element of the output vector of AUDst. The classification performance of SN should be evaluated along with knowledge distillation. The performance can be evaluated by comparing the hard prediction y^sthp of SN with the ground truth ys of the source domain data. y^sthp is obtained as follows:(10)y^st_hp=SoftmaxAUDstfst_s, θsti=eAUDstfst_s, θsti∑j=1KeAUDstfst_s, θstj
where y^sthp is a predicted classification label of xs in the SN and is used as a hard prediction value of SN. In Equation (10), the temperature parameter is not used.(2)Robust domain adaptation: domain adaptation is performed with robust features extracted with knowledge distillation to obtain domain-independent features. Domain adaptation consists of the following three stages: feature extractor, domain classifier, and appliance usage detection. First, a feature extractor FEst of SN is used. A feature extractor FEstxs, θfst of the source data and an FEstxt, θfst of the target data share a parameter set. Models learned with only source data are difficult to express with target data. To adapt the target domain data representation to FEst, the model learns the feature distribution difference between the two domains using MMD and minimizes it. The MMD distance is obtained as follows:(11)MMDXs,Xt=‖1ns∑i=1nsφxsi−1nt∑j=1ntφxtj‖H
where φ is a feature space mapping function that turns the original feature space into the reproducing kernel Hilbert space H. Further descriptions of the kernel are given in the following subsection. The domain classifier DCf, θdc learns by setting the ground truth values of the source domain data and the target domain data dcs=0 and dct=1, respectively, to separate the domain-independent features from the feature extractor. DCf, θdc has an output dc^s for source domain data and an output dc^t for target domain data. The two outputs are defined as follows:(12)dc^s=SoftmaxDCfst_s, θdc
(13)dc^t=SoftmaxDCfst_t, θdc
where fsts is the source domain feature, fstt is the target domain feature and θdc is the parameter set of DC. dc^s and dc^t values between 0 and 1. DC can obtain domain-independent features from FEst by learning that the source and target domains cannot be classified. Appliance usage detection uses the AUDst of SN. AUDst verifies classification performance using source data as input domain-independent features. The prediction of device usage detection can be obtained using Equation (10). In network inference, prediction of the target domain may be obtained using Equation (14).
(14)y^t=SoftmaxAUDstfst_t, θst
where y^t is the prediction of target data. Detection performance for target domain data is evaluated by comparing y^t with ground truth yt of target domain data.(3)Domain stabilization: The target domain data is pseudo-labeled with AUDst to enhance the data, thereby stabilizing the domain and improving the performance of the network. First, the feature fstt of the target domain data xt is input to the AUDst. If SoftmaxAUDstfstt, θst is obtained through Equation (14), PL is generated as a prediction value having the highest probability among Softmax values. However, if the probability is lower than the threshold, the data is not pseudo-labeled. The threshold is obtained experimentally. Domain stabilization consists of three steps, such as feature extraction and domain classifier. Appliance usage detection uses the following three types of data: source data (Xs, Ys), pseudo-labeled target data (Xt, Ytl), and unlabeled target data Xt. For feature extraction, fsts, fsttl and fstt are output through FEst. DC has no change in the domain, fsts, fsttl and fstt are classified as inputs, as in Equations (12) and (13). The appliance usage detection performs AUDstfsts,fsttl; θst. 

### 3.2. Network Losses

We carefully design network losses to obtain domain-independent features from feature distributions. We divide the network loss into the following four stages: knowledge distillation loss, feature distribution difference loss, domain classification loss, and appliance usage detection loss.

(1)Knowledge distillation loss: As shown in Figure 1, the knowledge distillation phase loss is the sum of the distillation loss $L_{ds}$ and student loss $L_{ds}$. $L_{ds}$ is to include the difference in the classification results of the TN and the SN in the loss. $L_{ds}$ is defined as follows:(15)Lds=2αT2LceeAUDtefte, θteiT∑j=1KeAUDtefte, θtejT,eAUDstfsts, θstiT∑j=1KeAUDstfsts, θstjT         =2αT2LceSoftmaxwith TAUDtefte, θtei,Softmaxwith TAUDstfsts, θsti                  =2αT2Lcey^te,y^st_sp
where $L_{ce}$ is the cross-entropy loss and α is the learning rate. The cross-entropy loss is calculated about teacher and student output. If the classification results of the teacher and the student are the same and distillation is good, $L_{ds}$ takes a small value. Additionally, $L_{st}$ means the cross-entropy loss of the classification of SN. $L_{st}$ is defined as follows:(16)$$\begin{array}{l} L_{st}=\left(1-\alpha \right)L_{ce}\left(\frac{e^{AUD_{st}{\left(f_{st_{s}},\;\theta_{st}\right)}_{i}}}{\sum_{j=1}^{K}e^{AUD_{st}{\left(f_{st_{s}},\;\theta_{st}\right)}_{j}}},y_{s}\right)=\left(1-\alpha \right)L_{ce}\left(Softmax\left(AUD_{st}{\left(f_{st_{s}},\;\theta_{st}\right)}_{i}\right),y_{s}\right)\\ \;=\left(1-\alpha \right)L_{ce}\left({\hat{y}}_{st\_hp},y_{s}\right) \end{array}$$Even in a network with relatively fewer parameters than in the TN, $L_{st}$ is also reduced when $L_{ds}$ is smaller, so it shows good feature extraction and classification performance.

(2)Feature distribution difference loss: As shown in Figure 1, the feature distribution difference loss is MMD Loss [44] $L_{f}$. $L_{f}$ estimates the difference between the feature distribution of the source domain data $X_{s}$ and the feature distribution of the target domain data $X_{t}$ through MMD. $L_{f}$ is generally defined as follows:(17)$$\begin{array}{l} L_{f}\left(f_{st\_s},f_{st\_t}\right)=MMD^{2}\left(f_{st\_s},f_{st\_t}\right)=\|E_{X_{s}\sim f_{st\_s}}\varphi \left(X_{s}\right)-E_{X_{t}\sim f_{st\_t}}\varphi {\left(X_{t}\right)\|}_{H}^{2}\\ \;=‹E_{X_{s}\sim f_{st_{s}}}\varphi \left(X_{s}\right),E_{{X^{\prime }}_{s}\sim f_{st_{s}}}\varphi {\left({X^{\prime }}_{s}\right)›}_{H}+‹E_{X_{t}\sim f_{st_{t}}}\varphi \left(X_{t}\right),E_{{X^{\prime }}_{t}\sim f_{st_{t}}}\varphi {\left({X^{\prime }}_{t}\right)›}_{H}\\ \;-2‹E_{X_{s}\sim f_{st\_s}}\varphi \left(X_{s}\right),E_{X_{t}\sim f_{st\_t}}\varphi {\left(X_{t}\right)›}_{H} \end{array}$$For the mapping function φ of Equation (17), we use kernel tricks because computational resources are required too much to obtain all the moments. We utilize the Gaussian kernel as shown in Equation (18).
(18)$$gk\left(x,y\right)=exp\left(-\frac{\|x-{y\|}^{2}}{2\sigma^{2}}\right)$$
where *g*k is the Gaussian kernel. In the Equation (18), Taylor’s development of the exponential develops as in Equation (19).
(19)$$e^{x}=1+x+\frac{1}{2!}x^{2}+\frac{1}{3!}x^{3}+\cdots$$Since Equation (19) contains all the moments for *x*, we use the Gaussian kernel. G$k\left(x,y\right)\;$is derived as Equation (20).
(20)$$gk\left(x,y\right)=‹\varphi \left(x\right),\varphi {\left(y\right)›}_{H}$$When Equation (15) is organized using Equation (20), $L_{f}$ is re-formulated as shown in Equation (21).
(21)$$\begin{array}{l} L_{f}\left(f_{st\_s},f_{st\_t}\right)=‹E_{X_{s}\sim f_{st\_s}}\varphi \left(X_{s}\right),E_{X\prime_{s}\sim f_{st\_s}}\varphi {\left(X\prime_{s}\right)›}_{H}+‹E_{X_{t}\sim f_{st\_t}}\varphi \left(X_{t}\right),E_{X\prime_{t}\sim f_{st\_t}}\varphi {\left(X\prime_{t}\right)›}_{H}\\ \;-2‹E_{X_{s}\sim f_{st_{s}}}\varphi \left(X_{s}\right),E_{X_{t}\sim f_{st_{t}}}\varphi {\left(X_{t}\right)›}_{H}\\ \;=E_{X_{s}{X^{\prime }}_{s}\sim f_{st_{s}}}gk\left(X_{s},{X^{\prime }}_{s}\right)+E_{X_{t}{X^{\prime }}_{t}\sim f_{st_{t}}}gk\left(X_{t},{X^{\prime }}_{t}\right)-2E_{X_{s}\sim f_{st\_s},X_{t}\sim f_{st\_t}}gk\left(X_{s},X_{t}\right) \end{array}$$

(3)Domain classification loss: As shown in Figure 1, the domain classification loss $L_{dc}$ is related to $FE_{st}$ and DC. $DC\left(f,\;\theta_{dc}\right)$ is modeled so that the source domain and the target domain cannot be distinguished. To minimize the distribution difference between $f_{{\mathrm{st}}_{s}}$ and $f_{{\mathrm{st}}_{t}}$, the loss of $DC\left(f,\;\theta_{dc}\right)$ should be maximized. Using ${\hat{dc}}_{s}$ and ${\hat{dc}}_{t}$ of $DC\left(f,\;\theta_{dc}\right)$, cross-entropy loss as a binary classifier-based $L_{dc}$ can be obtained as Equation (22).
(22)$$L_{dc}\left(x_{s},x_{t};\theta_{f\_st},\theta_{dc}\right)=-\overset{sn}{\underset{i=1}{\sum }}\left[\log\left(1-{\hat{dc}}_{s}^{i}\right)+\log\left({\hat{dc}}_{t}^{i}\right)\right]$$
where, sn is the sample number of mini-batch.

(4)Appliance usage detection loss: as shown in Figure 1, the appliance usage detection loss uses $L_{st}$ in the domain adaptation phase and $L_{aud}$ in the robust domain adaptation phase. Since both losses are applied to the same $\mathrm{AU}D_{st}$, the same loss equation is formularized as in Equations (23) and (24).
(23)$$L_{st}=L_{ce}\left(Softmax\left(AUD_{st}{\left(f_{st\_s},\;\theta_{st}\right)}_{i}\right),y_{s}\right)$$
(24)$$L_{aud}=L_{ce}\left(Softmax\left(AUD_{st}{\left(f_{st\_s},\;\theta_{st}\right)}_{i}\right),y_{s}\right)+L_{ce}\left(Softmax\left(AUD_{st}{\left(f_{st\_tl},\;\theta_{st}\right)}_{i}\right),y_{t}\right)$$Each neural network is learned by differentiating loss with corresponding weights, as shown in the dotted line in Figure 1.

### 3.3. Training Strategy

According to the network loss discussed above, the final optimization objective can be expressed as follows:(25)$$\left(\theta_{f\_st}^{*},\theta_{st}^{*},\theta_{dc}^{*}\right)=\arg\min\left[L_{aud}+L_{dc}+L_{f}\right]\;$$

Assuming that θf_te, θte are pre-learned high-performance networks, they do not perform additional learning to reduce network loss. When we learn Ldc of Equation (22), we apply the gradient reversal layer (GRL) to learn in a direction that fails to classify domains. The pseudo-code of the proposed model is summarized in Algorithm 1.
**Algorithm 1:** Parameter optimization procedure of the proposed method.**Input**: The source domain data xs,ys, The target domain data xt with M total samples, respectively.
**Output**: The optimized parameters θfst∗,θst∗,θdc∗**# Knowledge Distillation Phase****for** m = 0 to epochs **do****for** *n* to minibatch **do**  **#Foward propagation**  Teacher: fte←FEtexs, θf_te, y^te←AUDtefte, θte  Student: fst_s←FEstxs, θf_st, y^st_sp←AUDstfst_s, θst, y^st_hp←AUDstfst_s, θst  Lds←y^te,y^st_sp=2αT2Lcey^te,y^st_sp, Lst←y^st_hp,ys=1−αLcey^st_hp,ys

L←Lds+Lst  
**#Back propagation**  θf_st, θst←Adam∇θL, θf_st, θst  **end for****end for****# Domain Adaptation Phase****for** m = 0 to epochs **do****for** *n* to minibatch **do**  **#Foward propagation**  Source: fst_s←FEstxs, θf_st,  dc^s←DCfst_s, θdc,  y^st_hp←AUDstfst_s, θst   Target: fst_t←FEstxt, θf_st, dc^t←DCfst_t, θdcLf←fst_s,fst_t=EXsX′s~fstsgkXs,X′s+EXtX′t~fsttgkXt,X′t−2EXs~fsts,Xt~fsttgkXs,Xt, Ldc←xs,xt;θf_st,θdc=−∑i=1snlog1−dc^si+logdc^ti, Lst←fst_s, θst=LceSoftmaxAUDstfst_s, θst,ys L←Lf+Ldc+Lst   **#Back propagation**θf_st,θst, θdc←Adam∇θL, θfst, θst, θdc  **end for****end for****# Robust Domain Adaptation Phase****#Pseudo labeling**fst_t←FEstxt, θf_st, ytl← AUDstfst_t, θst**for** m = 0 to epochs **do****for** *n* to minibatch **do**  **#Foward propagation**  Source: fst_s←FEstxs, θf_st,  dc^s←DCfst_s, θdc,  y^st_hp←AUDstfst_s, θst   Target: fst_t←FEstxt, θf_st, dc^t←DCfst_t, θdc   Pseudo Target: fst_tl←FEstxt, θf_st,   y^st_tl←AUDstfst_tl, θst Lf←fst_s,fst_t=EXsX′s~fstsgkXs,X′s+EXtX′t~fsttgkXt,X′t−2EXs~fsts,Xt~fsttgkXs,Xt, Ldc←xs,xt;θf_st,θdc=−∑i=1snlog1−dc^si+logdc^ti, Laud←fsts,fsttl; θst       =LceSoftmaxAUDstfst_s, θsti,ys+LceSoftmaxAUDstfst_tl, θsti,ytl L←Lf+Ldc+Laud   **#Back propagation**   θf_st,θst, θdc←Adam∇θL, θfst, θst, θdc  **end for****end for**   θfst∗,θst∗,θdc∗


## 4. Experiments

### 4.1. Data Preparation

#### 4.1.1. Dataset

Two publicly available NILM datasets, UK-DALE [45] and REDD [46], were used for performance evaluation. UK-DALE collected smart meter data from five UK buildings, with sampling resolution and corresponding device-level consumption of 1 s and 6 s, respectively, for the total home consumption. The data set was recorded for 39–600 days. REDD was collected from six actual buildings in the United States. The measurement period is between 3 and 19 days, consisting of appliance-level energy consumption data sampled every 3 s and total measurements sampled every 1 s.

This article analyzes the use of the following five representative house appliances: dishwasher (DW), refrigerator (FG), kettle (KT), microwave (MV), and washing machine (WM). Since REDD does not have kettle data, NILM uses four house appliances, excluding kettles. The selected electronic products exhibit various power patterns and power levels. FG generally consumes constant and low power; however, other power consumption is very high power. DW and WM have very complex power usage patterns and power strengths. MV and KT have very monotonous power usage patterns. These five home appliances are generally designated as representative research targets because they account for most of the power consumption in the building.

In UK-DALE, House 1 uses data collected for 74 days from 1 January 2013 to 15 March 2013, and House 2 uses data collected for 74 days from 1 June 2013 to 13 August 2013. In REDD, House 1 and House 3 use data collected over 39 days from 17 April 2011 to 25 May 2011.

#### 4.1.2. Data Preprocessing

Each power consumption of the two datasets is downsampled to 1 min and then preprocessed for missing values using linear interpolation. Each house appliance is classified as ON (1) if the power consumption (for 15 min) is greater than the experimentally set threshold and is classified as OFF (0) if it is less than the threshold. Figure 3 and Figure 4 show the power usage of each home appliance in UK-DALE and REDD, respectively, and the thresholds for determining the ON event accordingly. The threshold was experimentally determined to be sure to include all ON states. However, since the FG is continuously operating, the threshold was determined based on the state in which the motor was running. Table 1 shows the exact threshold value of each home appliance and the number of ON events determined accordingly. The split ratio of training, validation, and test data are 6:2:2. The sliding window is used for around 15 min based on the ON event. A sliding window W with a stride length $l_{s}$ runs the sequence forward to obtain an input sample $x=\left(x_{1},x_{2},\ldots ,x_{W}\right)$. For each $i^{th}$ window, the network has $y^{i}=\left(y_{DW}^{i},y_{FG}^{i},y_{KT}^{i},y_{MV}^{i},y_{WM}^{i}\right)$ as output power.

### 4.2. Experimental Setup

#### 4.2.1. Implementation Configuration

To obtain an input sample, W is set to 15, and $l_{s}$ is set to 15 so that data is non-overlapped. In the TN, there are 3.2 times more parameters in the feature extractor and 1.6 times more parameters in the fully connected layer compared to the SN. The epochs in the robust domain adaptation and the domain stabilization phases are not set separately because the early stopping parameter automatically controls learning. The basic structure of SN is cited in [20]. The TN is experimentally determined to have a structure approximately twice as large as the SN. The mini-batch size is set to the maximum value applicable in the experimental environment. The decaying learning rate is used to determine the optimal value by repeatedly reducing it by one-third. The parameters of the proposed model are listed in Table 2.

All experimental models were modified and executed in Python 3.6 [47] and the Pytorch framework [48], and learning and inferencing used the NVIDIA RTX 2070 SUPER.

#### 4.2.2. Ablation Study Methods

Our model consists of the following four main techniques: TCN, gkMMD, teacher–student (TS) structure, and PL. We introduce an ablation study on five methods to investigate how individual components influence performance improvements in the proposed model.

Baseline: Typical domain adaptation method with BiLSTM-based feature extractors;TCN-DA: Domain adaptation method with TCN-based feature extractor;gkMMD-DA: Domain adaptation method with Gaussian kernel trick-based MMD Loss in baseline;TS-DA: A domain adaptation method for extracting features based on the robust knowledge distillation of the teacher–student structure. The feature extractor of SN used BiLSTM, such as the baseline, and the feature extractor of TN used BiLSTM, which is four times the size of the student;PL-DA: How to perform domain optimization with pseudo labeling on baseline method

#### 4.2.3. Evaluation Metrics

Performance evaluation uses the F1-score, a general metric. The F1-score is derived as shown in Equation (26).
(26)$$F1\left(TP,FP,FN\right)=\frac{2}{\frac{1}{\frac{TP}{TP+FP}}+\frac{1}{\frac{TP}{TP+FN}}}=\frac{2TP}{2TP+FP+FN}$$
where TP is true positive, FP is false positive, and FN is false negative.

To the best of our knowledge, there is no low sampling-based classification study in the domain adaptation field for NILM. Therefore, we did not conduct a one-on-one comparison with other studies.

### 4.3. Case Studies and Discussions

In this section, we conduct an experiment assuming two cases. In the first case, a house was designated as a source domain and a house was designated as a target domain within the same dataset. The second case was experimented with by specifying a source domain and a target domain between different datasets. Table 3, Table 4 and Table 5 show the F1 scores of domain adaptations for six segmentation methods. The ‘Improvement’ row shows how much the proposed method has improved. In addition, experiments on ablation studies are included, indicating how much each method affects overall performance.

#### 4.3.1. Domain Adaptation within the Same Dataset

In this subsection, experiments are carried out on the first case described above. In Table 3, $U^{1}$ denotes House1, $U^{2}$ denotes House2, $R^{1}$ denotes House1 of REDD, and $R^{3}$ denotes House3 of REDD. There is no result for the appliances since REDD does not have a kettle, and DW is not used in $R^{3}$.

Based on the baseline, TCN-DA was the method that had the most influence on performance except for our method, showing an average performance improvement of 3.38%. Next, TS-DA showed a performance improvement of 2.45%. In the case of gkMMD-DA, there was a bit of performance improvement or slightly reduced performance. Table 4 shows F1 score for TCN and gkMMD. gkMMD generally helps improve the performance when used with networks with residual blocks. PL-DA showed an average performance stabilization of 0.51% because it learns models in the direction of stabilizing the domain by finetuning the network. Our method showed a significant performance improvement of 6.03% on average compared to the baseline.

#### 4.3.2. Domain Adaptation between Different Datasets

In this subsection, experiments are performed on the second case described above. In Table 5, UK-DALE → REDD is an experiment using UK-DALE as a source domain and REDD as a target domain, and REDD → UK-DALE is an experiment using REDD as a source domain and UK-DALE as a target domain.

In the second case experiment, the average performance is improved by 5.74% even though the degree of domain characteristic change is greater than that of the first case experiment. Although the domains are different, the same type of appliance has almost the same pattern as the power usage, so the domain adaptation is well performed. Therefore, we have confirmed the possibility that in the field of NILM, we do not have to learn new neural networks even if each household and living area are different. Our method shows better results compared to the baseline.

Experiments show that domain adaptations within the same dataset perform well when the proposed method is used, and performance improvements can also be seen for domain adaptations between different datasets. It is a very significant result that our method without individual model learning for all households achieves a performance improvement of 5–6% through only one learning. There are several main reasons for improving accuracy. (1) Rich domain independent feature information is extracted by learning through teacher–student-based knowledge distillation. (2) By using TCN residual blocks and gkMMD together can effectively reduce the distribution mismatch between the two domains. (3) PL can stabilize the network’s decision boundaries.

#### 4.3.3. Discussions

The proposed model can automatically track the use of individual appliances under full load. We look at a series of our method-based applications for elderly households living alone and public electricity management institutions.

In the case of elderly households living alone, the risk of dying alone is generally very high. This risk situation is one of the critical problems to be solved at the government level. By analyzing device usage patterns, it is possible to develop a household risk detection system through abnormality detection in the household. Efficient energy management is an essential issue in public electricity management institutions. It is possible to develop an energy management system that adjusts the power generation ratio by identifying and managing energy-inefficient customers using home appliance usage patterns and power usage.

There are several limitations to the proposed method. (1) Domain adaptation is difficult to apply if house appliances of source and target data are different. (2) The difference in power usage between households is so large that the data imbalance is severe. (3) Although performance is improved by reducing distribution differences over the source and target features, there is no clear academic basis for extracting domain-independent features by reducing distribution differences. It is generally on an experimental basis. In future work, we aim to address the second limitation, which is the data imbalance. Data imbalance is the most fundamental problem in neural network training. Future work is planned in the direction of GAN-based sampling methods to resolve data imbalance or networks that perform high-quality learning despite data imbalance.

## 5. Conclusions

We developed a novel methodology that combines robust knowledge transfer and network stabilization for NILM to improve previous tasks and perform generalization across domains. The proposed method improves the detection performance of device usage for unlabeled target domain data by using a network trained only on the labeled source data. Teacher–student-based knowledge distillation is adopted to transfer quality features from the source domain. PL is utilized for domain stabilization through low-density separation between classes and entropy regularization effects. gkMMD is employed to reduce distribution differences between domain-independent features. Based on various techniques, we improve the performance of the proposed domain adaptation method by considering the distribution of robust domain-independent features.

To prove the proposed method, we used UK-DALE data and REDD as data. For data preprocessing, data such as training, verification, and testing were constructed by experimentally setting thresholds for distinguishing ON events in each appliance. Five methods of ablation study were performed for the performance test. Within the same dataset, domain adaptation improved the F1 score of the proposed method over the baseline by an average of 6.04%. Domain adaptation on different datasets improved the F1 score of the proposed method over the baseline by an average of 5.74%. While performance has not improved significantly for problems with much larger domain feature changes, maintaining existing performance alone is a great achievement.

## Figures and Tables

**Figure 1 sensors-22-05838-f001:**
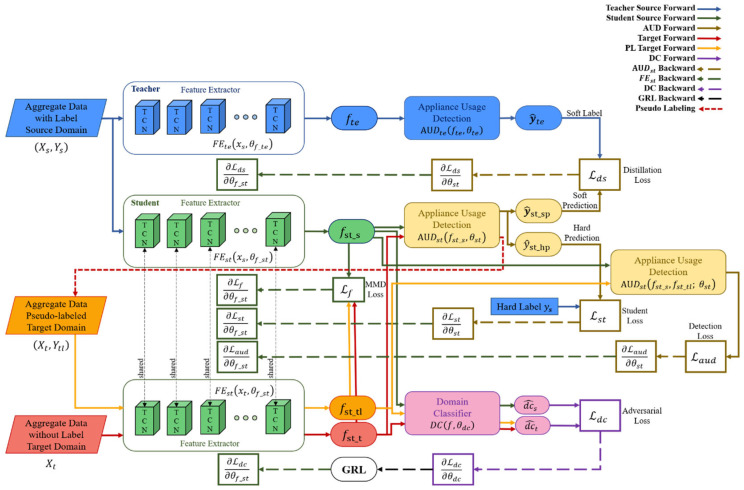
A detailed overall configuration diagram of the proposed semi-supervised domain adaptation for multi-label classification on nonintrusive load monitoring.

**Figure 2 sensors-22-05838-f002:**
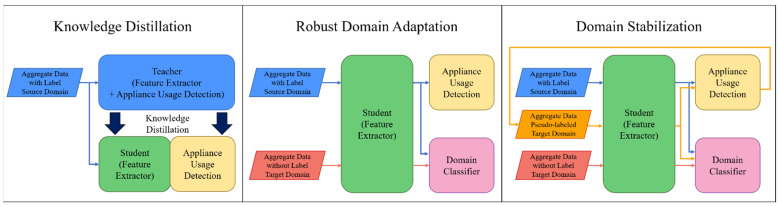
Step-by-step flowchart of the proposed method.

**Figure 3 sensors-22-05838-f003:**
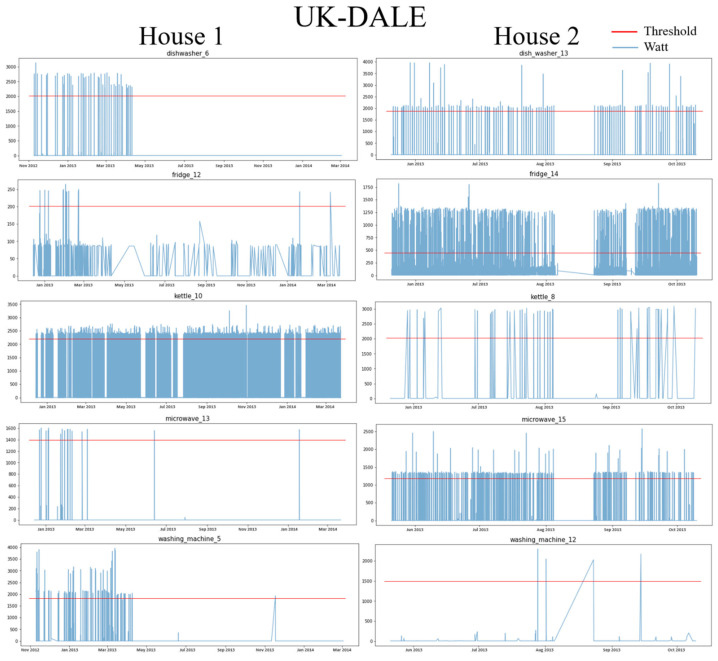
Power Usage and ON Thresholds for House 1 and House 2 in the UK-DALE dataset.

**Figure 4 sensors-22-05838-f004:**
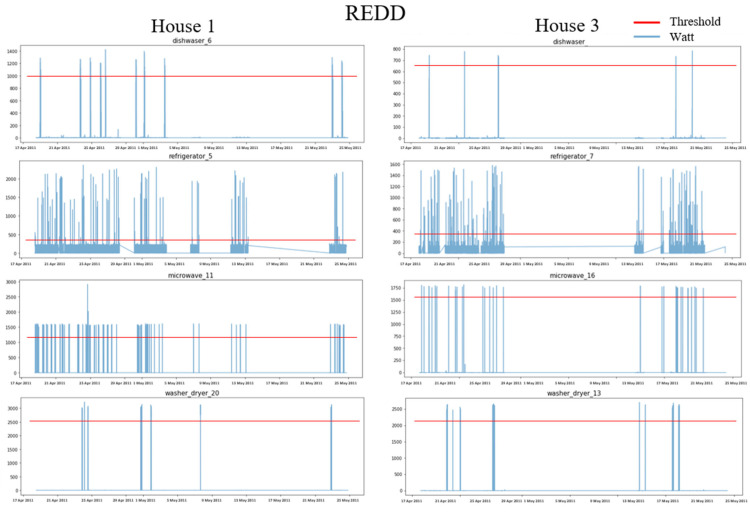
Power usage and ON thresholds for House 1 and House 3 in the REDD dataset.

**Table 1 sensors-22-05838-t001:** ON threshold and the number of ON events in UK-DALE and REDD datasets.

	UK-DALE				REDD			
	House 1		House 2		House 1		House 3	
Appliance	Threshold	The Number of ON Event	Threshold	The Number of ON Event	Threshold	The Number of ON Event	Threshold	The Number of ON Event
DW	2000	4431	1800	3236	1000	6712	650	2934
FG	250	2441	400	5291	400	2944	350	3344
KT	2200	4495	2000	1694	-	-	-	-
MV	1400	1242	1200	4218	1200	4809	1600	1327
WM	1800	4980	1500	1524	2500	4796	2200	5764

**Table 2 sensors-22-05838-t002:** Training parameters.

Parameter Description	Value
Number of TCN blocks	8 (TN)
	5 (SN)
Number of filters in each TCN block	128 (TN)
	64 (SN)
Filter size	3
Number of fully connected layers	5 (TN)
	3 (SN)
	2 (Domain Classifier)
Dilation factor	$2^{i}$ for block i
Activation function	ReLU
Dropout probability	0.1
Number of maximum epochs	200
Number of minimum early stopping epochs	4
Mini-batch size	512
Learning rate	3 × 10^−3^

**Table 3 sensors-22-05838-t003:** F1 score comparison of domain adaptation within the same dataset.

Appliance	Method	UK-DALE		REDD	
		$\left(U^{1}\to U^{2}\right)$	$\left(U^{2}\to U^{1}\right)$	$\left(R^{1}\to R^{3}\right)$	$\left(R^{3}\to R^{1}\right)$
DW	Baseline	0.781	0.805	−	−
	TCN-DA	0.832	0.827	−	−
	gkMMD-DA	0.778	0.793	−	−
	TS-DA	0.812	0.826	−	−
	PL-DA	0.787	0.811	−	−
	Ours	0.822	0.832	−	−
	Improvement	5.25%	3.35%	−	−
FG	Baseline	0.833	0.834	0.817	0.818
	TCN-DA	0.842	0.841	0.829	0.840
	gkMMD-DA	0.837	0.836	0.819	0.819
	TS-DA	0.850	0.853	0.824	0.827
	PL-DA	0.834	0.845	0.818	0.819
	Ours	0.875	0.872	0.843	0.852
	Improvement	5.04%	4.56%	3.18%	4.16%
KT	Baseline	0.761	0.832	−	−
	TCN-DA	0.811	0.839	−	−
	gkMMD-DA	0.753	0.820	−	−
	TS-DA	0.807	0.835	−	−
	PL-DA	0.770	0.833	−	−
	Ours	0.817	0.868	−	−
	Improvement	7.36%	4.33%	−	−
MV	Baseline	0.742	0.791	0.793	0.790
	TCN-DA	0.751	0.798	0.806	0.721
	gkMMD-DA	0.746	0.795	0.797	0.774
	TS-DA	0.753	0.803	0.804	0.798
	PL-DA	0.744	0.796	0.794	0.793
	Ours	0.774	0.812	0.814	0.818
	Improvement	4.31%	2.65%	2.65%	3.54%
WM	Baseline	0.615	0.611	0.841	0.782
	TCN-DA	0.725	0.708	0.844	0.799
	gkMMD-DA	0.623	0.625	0.842	0.786
	TS-DA	0.668	0.653	0.832	0.783
	PL-DA	0.623	0.615	0.843	0.783
	Ours	0.736	0.713	0.870	0.832
	Improvement	19.67%	16.69%	3.45%	6.39%

**Table 4 sensors-22-05838-t004:** F1 score comparison of TCN + gkMMD domain adaptation within the same dataset.

Appliance	UK-DALE		REDD	
	$\left(U^{1}\to U^{2}\right)$	$\left(U^{2}\to U^{1}\right)$	$\left(R^{1}\to R^{3}\right)$	$\left(R^{3}\to R^{1}\right)$
DW	0.823	0.828	−	−
FG	0.857	0.854	0.834	0.847
KT	0.813	0.841	−	−
MV	0.762	0.805	0.809	0.764
WM	0.730	0.709	0.852	0.815

**Table 5 sensors-22-05838-t005:** F1 score comparison of domain adaptation between different datasets.

Appliance	Method	UK-DALE →REDD	REDD → UK-DALE
DW	Baseline	0.741	0.712
	TCN-DA	0.779	0.737
	gkMMD-DA	0.736	0.713
	TS-DA	0.770	0.745
	PL-DA	0.747	0.714
	Ours	0.778	0.747
	Improvement	4.99%	4.92%
FG	Baseline	0.786	0.764
	TCN-DA	0.794	0.787
	gkMMD-DA	0.787	0.769
	TS-DA	0.800	0.772
	PL-DA	0.787	0.770
	Ours	0.821	0.797
	Improvement	4.45%	4.32%
MV	Baseline	0.719	0.739
	TCN-DA	0.726	0.716
	gkMMD-DA	0.719	0.746
	TS-DA	0.729	0.749
	PL-DA	0.717	0.743
	Ours	0.742	0.763
	Improvement	3.2%	3.25%
WM	Baseline	0.563	0.758
	TCN-DA	0.669	0.773
	gkMMD-DA	0.573	0.766
	TS-DA	0.610	0.758
	PL-DA	0.568	0.763
	Ours	0.672	0.769
	Improvement	19.36%	1.45%

## Data Availability

REDD and UK-DALE datasets can be found at http://redd.csail.mit.edu/ (accessed on 30 May 2022) and https://ukerc.rl.ac.uk/DC (accessed on 30 May 2022), respectively.

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
