# Peer review of "Semi-Supervised Domain Adaptation for Multi-Label Classification on Nonintrusive Load Monitoring"

_sensors, 2022, doi:10.3390/s22155838_

Round 1

Reviewer 1 Report

This paper presents a classification method in the field of non-intrusive load monitoring (NILM), which combines the robust feature extraction method based on the Teacher-Student structure and the domain stabilization method based on pseudo-labeling. Meanwhile, the research uses the domain adaptability research method, combines a large number of experimental data, and compares the performance of various algorithms. The overall logic of the article is strong, the structure is complete, and the research content is relatively sufficient and reasonable.

However, there are some deficiencies in my opinion:

(1) The expression of concepts and tools in the research field is not clear enough, and some abbreviations do not indicate the exact meaning when they appear for the first time.

(2) The introduction of relevant work in this paper is too straightforward, only the content and basic principles of relevant work are introduced, and there are few summary words. The common methods, progress, characteristics and areas for improvement of relevant work are rarely mentioned.

(3) The pictures and tables are clear, but a little bloated.

(4) These numerous descriptions of the principles of existing methods and models are very clear, but a little too much.

(5) The language expression in the fourth section is not academic enough.

(6) The second person appears in the fourth section, and the written expression is not academic enough.

(7) The study on the combination of NILM and domain adaptability is innovative.

Author Response

Thank you for taking your precious time to review the quality. Please refer to the attached file.

Reviewer 2 Report

The conclusions have several claims of improvement, however, none of this improvement is quantified and neither supported by the experimental data that was provided. 
Use of language requires significant improvement, in syntax, grammar, and even the correct use of some words in sentences where the understanding of the sentence is compromised by incorrect word usage.
All symbols used should be explained. I suggest the use of figure legends.

Author Response

(The authors gave the same response as above.)

Reviewer 3 Report

The manuscript is well structured and well-argued. However, several rectifications and modifications are required to ensure that its quality stands up to this reputed journal.  

1.       The authors have carried out the study based on simulation/experimentation to improve the Multi-label classification on non-intrusive load monitoring using a methodology that combines robust knowledge transfer and network stabilization, which is the strong point of the manuscript.

2.        The English language must be improved. There are several grammatical errors as one goes through the manuscript that requires rectification. Most of the sentences convey no proper meaning and could be off-putting to the readers and practitioners.

3.       The first section introduces a basic outlook on importance of optimized energy usage management and non-intrusive load monitoring (NILM) and also a brief literature review of performance of NILM technologies is presented.

4.       Suggested to give the details of data set size, percentage of data used for training, validation and testing by the models used for classification study in semi-supervised domain adaptation. More detailed explanation of training parameters mentioned in Table-1 is needed.

5.       The presentation of mathematical equations can be improved and some equations are not numbered.

6.       How many features are selected for study? Give the number of samples observed from each data set. What factors affect the selection of learning rate of proposed model?

7.       What are the alternative quantitative and qualitative parameters can be considered to evaluate the performance of domain adaptation method other than F1 score?

8.       The manuscript has the potential to be improved and requires major rectifications. With that being said, I wish the authors all the best on their endeavor to improve the quality of the manuscript.

Author Response

(The authors gave the same response as above.)

Round 2

Reviewer 2 Report

There was a noticeable improvement from the original version. However, important aspects are still lacking in the submitted manuscript:
- The conclusions claim that the proposed method improved performance over some of the existing methods. Section 4.3 indicates improvements limited to 4.13%, and in many situations the improvement was below 1%. However, the claimed value of 6.03% if improvement is not shown in any of the tables presented in Section 4.3.
- Section 4.2.2 lists five methods used for comparison to the proposed methods:
- why were these specific methods used for comparison ?
- why aren't there any references to these methods ?
- was the implementation of these methods done by the authors of this paper, or are the results presented in Section 4.3 published in the literature by others ?
- if the authors implemented these five methods, how can I be sure that their implementation was optimal ? if not optimal, how could it be used for a comparison with such small margins as 0.4% to 4.13%?

Author Response

Thank you for taking your precious time to review. Please refer to the attached file.
